# Cognitive Performance during the Development of Diabetes in the Zucker Diabetic Fatty Rat

**DOI:** 10.3390/cells12202463

**Published:** 2023-10-16

**Authors:** Marcia Spoelder, Yami Bright, Martine C. Morrison, Veerle van Kempen, Lilian de Groodt, Malvina Begalli, Nikita Schuijt, Eva Kruiger, Ronald Bulthuis, Gabriele Gross, Robert Kleemann, Janna A. van Diepen, Judith R. Homberg

**Affiliations:** 1Department of Cognitive Neuroscience, Donders Institute for Brain, Cognition, and Behaviour, Radboud University Medical Center, Heyendaalseweg 135, 6525 AJ Nijmegen, The Netherlands; bright.yami@gmail.com (Y.B.);; 2Department of Metabolic Health Research, Netherlands Organisation for Applied Scientific Research (TNO), Sylviusweg 71, 2333 CE Leiden, The Netherlands; 3Metris B.V., Kruisweg 829c, 2132 NG Hoofddorp, The Netherlands; 4Medical and Scientific Affairs, Reckitt|Mead Johnson Nutrition Institute, Middenkampweg 2, 6545 CJ Nijmegen, The Netherlands

**Keywords:** diabetes, adolescence, cognition, insulin, inflammation

## Abstract

Increased insulin levels may support the development of neural circuits involved in cognition, while chronic mild inflammation may also result in cognitive impairment. This study aimed to gain more insight into whether cognition is already impacted during adolescence in a genetic rat model for obesity and type 2 diabetes. Visual discrimination learning throughout adolescence and the level of motivation during early adulthood were investigated in Zucker Diabetic Fatty (ZDF) obese and ZDF lean rats using operant touchscreens. Blood glucose, insulin, and lipids were longitudinally analyzed. Histological analyses were performed in the liver, white adipose tissues, and the prefrontal cortex. Prior to the experiments with the genetic ZDF research model, all experimental assays were performed in two groups of outbred Long Evans rats to investigate the effect of different feeding circumstances. Adolescent ZDF obese rats outperformed ZDF lean rats on visual discrimination performance. During the longitudinal cognitive testing period, insulin levels sharply increased over weeks in ZDF obese rats and were significantly enhanced from 6 weeks of age onwards. Early signs of liver steatosis and enlarged adipocytes in white adipose tissue were observed in early adult ZDF obese rats. Histological analyses in early adulthood showed no group differences in the number of prefrontal cortex neurons and microglia, nor PSD95 and SIRT1 mRNA expression levels. Together, our data show that adolescent ZDF obese rats even display enhanced cognition despite their early diabetic profile.

## 1. Introduction

Obesity is a chronic health condition that has been ranked as the fifth most common cause of mortality worldwide [1]. Genetic heritability and the obesogenic environment play important roles as causal pathways for obesity [2]. Inflammatory reactions in various critical organs, including the brain, have been associated with the obesogenic condition [3,4,5]. Moreover, neuroinflammation may result in reduced neuroproliferation in several brain areas, such as the prefrontal cortex and hippocampus [6,7,8,9,10,11]. Experimental studies with rodent models of insulin resistance, type 2 diabetes, or dietary obesity models have demonstrated deficits in a wide range of cognitive tasks [12,13,14]. However, these experiments were mainly performed in adult rodents upon long exposure to either the genetic or dietary obesity/diabetic model.

Cognitive decline is putatively due to neuronal signaling impairment, decreased synaptic plasticity, and reduced insulin signaling [15]. Interestingly, insulin itself plays a profound role in the formation of neural circuits and synaptic connections in rodents, non-human primates, and humans [16,17]. Several studies have convincingly shown that insulin administration enhances behavioral performances [18], protects against neuroinflammation [19], and inhibits pro-inflammatory factors in obese non-diabetic human subjects [20]. Consequently, decreased sensitivity to insulin caused by insulin resistance due to an obesogenic or diabetic condition and the resulting interruption of its signal transduction can challenge the positive effect insulin seems to have on brain functioning [12,16]. 

This study aimed to gain insight into the interplay between cognition and the development of obesity/diabetes during the adolescent period. We were interested in whether the enhanced levels of insulin during development had effects on cognition at an age where the consequences of low-grade systemic inflammation due to the production of inflammatory mediators by dysfunctional adipose tissue are slowly starting to emerge. Therefore, the obese Zucker Diabetic Fatty (ZDF) rat was selected since the homozygotes ZDF rats (Leprfa/Leprfa) spontaneously develop obesity, hyperlipidemia, insulin resistance, and hyperglycemia by 6 to 8 weeks of age [21,22]. Prior to the experiments with the obese ZDF research model, we performed all experimental assays in two groups of outbred Long Evans rats to investigate under which feeding circumstances (ad libitum food and sucrose rewards versus food restriction and grain rewards) the rats could acquire the full operant higher-order cognitive task within the adolescent period. We used a touchscreen-based task, which resembles those used to assess cognition in humans and thereby has a high translational impact [23,24]. We chose the visual discrimination task since we expected the rats to be able to acquire this long-term memory test within the full adolescent period. We hypothesized that when obese ZDF animals age and the physiological obesogenic phenotype emerges, their cognitive performances would decline over test sessions and that, overall, a lower cognition would be observed.

## 2. Materials and Methods

### 2.1. Animals and Dietary Regime

A total of 36 (18 males and 18 females) ZDF rats from Charles River (France) and 31 (15 males and 16 females) Long Evans (LE) rats from Janvier Labs (France) were used. Upon arrival at 3.5 weeks of age, animals were first socially housed in groups of four in standard Macrolon^®^, 55 × 33 × 20 cm cages for 5 days and thereafter were housed individually in Macrolon^®^, 42 × 26 × 20 cm cages. Animals were housed in a controlled environment (reversed 12/12 light-dark-cycle where white lights went off at 7.00 a.m., 22 °C) with water provided ad libitum. All experiments were performed in accordance with the European Parliament and Council Directive (2010/63/EU) and approved by the Centrale Commissie Dierproeven in The Hague, The Netherlands. ZDF rats were fed with a purified diet (AIN-93G, SSNIFF diet, Bio Services B.V., Uden, The Netherlands), and LE rats were fed with standard laboratory chow (V1534-703, SSNIFF diet, Soest, Germany). To motivate animals for repeated complex cognitive testing, two main feeding regime strategies were employed in LE rats: 1. ad libitum feeding and sucrose pellets [5TUT] as a reward, i.e., the LE-sucrose group, and 2. food restriction and regular grain pellets (Tablet [5TUM] 45 mg/pellet, TestDiet, USA, Richmond, Indiana) as a reward, i.e., the LE-grain group. Based on the results in LE rats, the food restriction, combined with the grain pellets reward, was chosen as the feeding regime for the ZDF rat model. The LE-sucrose group received ad libitum feeding. The other three groups, i.e., the LE-grain group and the lean and obese ZDF groups, were food-restricted up to 90% of free-feeding body weight during weekdays. On Friday afternoon, rats received an excess amount for the test-free weekend. On Monday mornings, the remaining food was weighed and used to determine how much food each individual rat consumed under ad libitum conditions. This amount was used to calculate the 90% dietary food restriction regime for the upcoming week. The animals were fed daily after cognitive testing. 

### 2.2. Behavioural Assessment


*General Approach*


In short, the rats were individually placed in LaborasTM cages for 48 h sessions at weeks 4, 6, 8, 10, and 12 of age to measure locomotor activity, eating, and drinking behavior. The operant behavior was assessed in touchscreen operant chambers (Bussey Rat Touchscreen Chamber, Campden Instruments, Loughborough, Leics, UK), whereby the pre-training and visual discrimination (VD) tests were performed during the adolescent period of the rat (~week 4 to 10 of age) and the motivational tests were performed during either adolescence or early adulthood, depending on the acquisition and performance speed during daily testing for each individual rat. The rats were tested twice a day for 30 min sessions, with a minimum interval of 3 h between sessions for all tests, except for the 60 min progressive ratio task, which was performed once per day. In the VD task, the rat is required to discriminate two images and touch the correct image in order to receive a reward pellet [25,26]. This task measures learning by stimulus discrimination and long-term memory while assessing the performances over sessions across several days [25]. 

#### 2.2.1. Phenotypic Behavioural Assessment

The Laboratory Animal Behavior Observation, Registration, and Analysis System (LABORAS, Metris) possesses transducers that can measure vibrations due to the movement of the rat on the platform. The software can subsequently convert the signals into behavioral parameters, such as locomotion, grooming, eating, and drinking. Additionally, the position of the animal can be assessed, and thereby, the traveled distance, the maximum speed, and other motion parameters were measured [27]. The locomotor activity, eating, and drinking behaviors from 5 consecutive measurements at weeks 4, 6, 8, 10, and 12 were averaged.

#### 2.2.2. Touchscreen Operant Chambers

Touchscreen chambers equipped with a touchscreen, infrared beams, white house light, pellet dispenser, and a feeder magazine with magazine light were placed within a sound-attenuating chamber, fitted with a camera positioned above the operant chamber to monitor the rats (Bussey Rat Touchscreen Chamber, Campden Instruments, Loughborough, Leics, UK). To register responses, the operant chambers were equipped with infrared beams inside the feeder magazine and on the surface of the touchscreen. A black mask with two apertures was placed in front of the touchscreen to minimize unintended responses on the touchscreen and to guide the rat towards the correct locations on the screen. Presentation of the stimuli, reward delivery, and data collection were conducted using ABET II Touch Software (Campden Instruments, Loughborough, Leics, UK) and a multimedia research control Whisker system [28].

#### 2.2.3. Training Stages

One week after arrival, between 3 and 4 weeks of age, rats started their cognitive training in the operant chambers. Prior to response training, a handful of reward pellets was provided to the rats in their homecage to reduce potential food neophobia. In the first training step, called habituation, the rats were introduced to the chamber and were provided with 25 reward pellets randomly delivered into the feeder magazine over the course of 30 min. The second training step consisted of autoshaping (AT), during which a conditioned stimulus (white circle, diameter 8 cm) on the right side of the touchscreen was introduced and associated with reward delivery. In this step, the rat received the reward either upon touching the image or when the 30 s stimulus presentation passed independently from whether the rat touched the screen or not. Each 30 min session consisted of 50 trials, and to continue to the next training stage, certain criteria had to be met (Table 1). After a correct response, one reward pellet was provided, the stimulus disappeared, and an inter-trial interval of 5 s followed. In the third step, defined as must touch (MT), rats were required to press the circular stimulus on the screen to obtain the reward, and a new trial would start only after the pellet was collected from the feeder magazine. An incorrect response or no response (omission) resulted in no reward delivery. In the next step, called punish incorrect (PI), a negative reinforcement was introduced. When the rat pressed the screen on the wrong location or in the absence of any stimulus presented, the house light turned on (illumination of the house light for 10 s), and no pellet was delivered. The last step of the training stages consisted of what is defined as moving punish incorrect (MPI), during which the stimulus changed position within the left and right aperture of the mask in random order.

#### 2.2.4. Visual Discrimination Task

After the acquisition training during the pre-training stages, the rats directly moved on to the visual discrimination (VD) task. In this task, the rat is presented for the first time with two images, one in each aperture of the mask. The images were horizontal white lines versus vertical white lines (4.5 cm × 4.5 cm) on a black background (10 × 10 cm). Each rat was allocated either the horizontal or vertical image as the correct image. The images randomly changed position on the touchscreen. The rewarded image (horizontal or vertical stripes) was counterbalanced between the rats. A correct response resulted in the direct removal of the stimulus, the delivery of one pellet, and, subsequently, an inter-trial interval. An incorrect response was followed by the illumination of the house light (punishment), no pellet delivery, and a correction trial started after 15 s. In a correction trial, the rewarded stimulus was presented at the same location as in the previous trial, and this continued until the rat made a correct response. When the rat did not make a response within 30 s (omission), a new trial started after 10s. The rats were tested for at least 10 sessions on the visual discrimination task twice a day, with a minimum interval of 3 h between the two daily sessions. Data acquisition within the visual discrimination task was completed upon the achievement of an average of at least 70% correct responses (correct responses/correct responses + incorrect responses) for three subsequent sessions with less than 15% variation in the percentage of correct responses among those selected three sessions. 

#### 2.2.5. Motivation Task

When rats successfully learned the VD, they were subjected to a fixed ratio (FR1, FR2, and FR5) and, subsequently, a progressive ratio (PR) schedule of reinforcement to assess the motivation to work for a reward. In these tasks, a white circle was shown on the right side of the screen that had to be touched 1, 2, 5, or a progressive number of times to receive the reward. In FR1, only a single response on the white circle was needed to receive a reward, comparable to the must-touch training stage. The rats were between 9 and 13 weeks of age during the FR sessions. FR1 was followed by FR2 and FR5, where the rat needed to press the white circle two and five times, respectively, to receive the reward. The stimulus remained present until the rat touched it for the correct number of times. After receiving the reward, the stimulus was removed from the screen, and a new trial began 5 s after reward collection. Between the removal of the stimulus on the screen and the reward collection, and between the reward collection and the new stimulus presentation, inter-trial interval responses were detected to assess perseverance in responding. For FR1, FR2, and FR5, the passing criteria were to obtain 50 rewards per session. Thus, the rats had to touch the stimulus in total, either 50, 100, or 250 times, respectively. When rats were successful in the FR tasks, they advanced to the PR schedule. A PR schedule (between weeks 10 and 15 of age) of reinforcement measures reward strength and has become the golden standard task to assess motivation in touchscreen-based tasks in behavioral neuroscience [29]. Within this task, the rat needs to increase the number of responses on a linear basis of 4 (i.e., 1, 5, 9, 13, etc.) to earn a reward. All rats were required to complete 10 PR sessions for ZDF rats and 20 PR sessions for LE rats. In 3 PR sessions, the rats received different reward pellets to assess whether there was a difference in motivation between grain and sucrose pellets as rewards. The LE-sucrose group received sucrose pellets during all cognitive testing except for 3 PR sessions (16, 18, 20), during which they received grain pellets. The LE-grain group received grain pellets during all cognitive testing except for 3 PR sessions (16, 18, 20), during which they received sucrose pellets. The ZDF rats were rewarded with grain pellets except for 3 PR sessions (6, 8, 10), during which ZDF rats were rewarded with sucrose pellets. The average of these three sessions with the new reward type was compared to the average of the three sessions of one session before the challenge sessions with the new reward type (session 15-17-19 for ZDF rats, or session 5-7-9 for LE rats). 

### 2.3. Physiological Assessment


*General Approach*


In short, blood glucose and plasma insulin measurements were taken in the blood that was collected via a tail puncture or tail cut in EDTA-coated tubes every two weeks for five times in total starting from week 4 of age. Lipid and Lipoprotein Analysis of total plasma cholesterol and triglycerides were measured in week 12 of age. The rats were sacrificed in early adulthood, around week 14 and week 15 of age for ZDF rats and between weeks 14 and 19 of age for LE rats. After transcardial perfusion, the left hemisphere was post-fixed in 2% paraformaldehyde and, 48 h thereafter, transferred into a 30% sucrose solution for a total of 7 days at 4 °C and finally stored at −80 °C. The right medial prefrontal cortex and hippocampus were dissected and collected in Eppendorf tubes and stored at −80 °C. A part of the liver, mesenteric, and perigonadal white adipose tissue were post-fixed and embedded in paraffin. Adipocyte morphometry [30] and pathological hallmarks in the liver were scored as detailed previously [31,32]. The number of neurons and microglia and the soma cell size area of microglia were visualized using immunohistochemistry. Gene expression analyses were performed for primers Rattus norvegicus Ywhaz, β-actin and Gapdh, SIRT1, and PSD-95, and were synthesized by Sigma (for primer pair sequences, see Table 2). Hippocampal insulin levels were measured via standard protein concentrations of the homogenized tissue and subsequent insulin ELISA.

#### 2.3.1. Blood Glucose and Plasma Insulin Measurements

The blood collection was performed on Mondays after providing ad libitum food during the weekends. Five hours prior to the blood collection, food was removed. Blood glucose measurements were directly performed with the glucometer Freestyle Lite (Abbott Diabetes Care, Hoofddorp, The Netherlands). Blood was centrifuged at 4000 rpm and 4 °C for 10 min, and plasma was collected and stored at −80 °C until further use. Plasma insulin levels were quantified using an Enzyme-Linked Immunosorbent Assay (ELISA) (Ultra-Sensitive Rat insulin ELISA, Crystal Chem Inc., Zaandam, The Netherlands), using 5 μL plasma in duplicate for each sample and executed according to the manufacturer’s instructions. The absorbance values were obtained with a 450 nm excitation filter and corrected with the absorbance value obtained from both a 630 nm excitation filter and the sample diluent. The corrected and averaged concentrations of 5 measurements (from week 4 to week 12) are presented as nanograms per milliliter.

#### 2.3.2. Lipid and Lipoprotein Analysis

Total plasma cholesterol (very low-density lipoprotein; VLDL-C, low-density lipoprotein; LDL-C and high-density lipoprotein; HDL-C) and plasma triglycerides were measured after 5 h fasting in week 12 of age using commercially available enzymatic assays (CHOD-PAP and GPO-PAP, respectively; Roche Diagnostics, Almere, The Netherlands). Plasma HDL-cholesterol was measured after precipitation of the apoB-containing lipoproteins (VLDL and LDL) from 20 μL EDTA plasma by adding 10 μL 0.2 M MnCl2 and 10 μL heparin (LEO Pharma, Amsterdam, The Netherlands; 500 U/mL). Mixtures were incubated for 20 min at room temperature and centrifuged for 15 min at 13,000 rpm at 4 °C to precipitate VLDL and LDL particles. The remaining cholesterol in the supernatant (HDL-C) was quantified with the aforementioned assay (CHOD-PAP). Furthermore, for ZDF rats, lipoprotein profiles were obtained by using the AKTA-fast protein liquid chromatography system (Pharmacia, Roosendaal, The Netherlands), as described previously [31]. The plasma from ZDF or lean rats for both sexes was pooled and analyzed, and the averaged profile of 24 fractions resulting from 2 analyses was presented. 

#### 2.3.3. Sacrifice and Tissue Collection

Rats were overdosed with approximately 2 mL pentobarbital (60 mg/mL) via an intraperitoneal injection. After sedation, blood was collected through cardiac puncture. After transcardial perfusion with 0.1 M phosphate-buffered saline (pH 7.2), rats were decapitated. The left hemisphere was post-fixed in 2% paraformaldehyde and, 48 h thereafter, transferred into a 30% sucrose solution for a total of 7 days at 4 °C. The sucrose solution was refreshed after 4 days. Thereafter, brains were stored at −80 °C until use for histological experiments. The medial prefrontal cortex was dissected and collected in Eppendorf tubes and stored at −80 °C until further use. The liver mesenteric and perigonadal white adipose tissues were weighted. A part of the liver, mesenteric, and perigonadal white adipose tissue was post-fixed in 4% paraformaldehyde for 48 h. These tissues were subsequently transferred to 70% ethanol in demi water for a total of 7 days at 4 °C. The ethanol solution was refreshed after 4 days. The liver, mesenteric, and perigonadal white adipose tissue parts were embedded in paraffin for later histological experiments.

#### 2.3.4. White Adipose Tissue and Liver Histology

Paraffin-embedded cross sections of adipose tissues (5 µm) and liver (3 µm) were stained with hematoxylin and eosin. Adipose tissue cross-sections were digitized with a slide scanner (Aperio AT2, Leica Biosystems, Amsterdam, The Netherlands), and adipocyte morphometry (average adipocyte size and adipocyte size distribution) was analyzed using an automated image analysis as described previously [32]. Liver cross sections were scored blindly by a board-certified pathologist using a grading method for human non-alcoholic steatohepatitis adapted for rodent models [33], and pathological hallmarks were scored as detailed previously [33]. In brief, macrovesicular steatosis, microvesicular steatosis, and hepatocellular hypertrophy were determined as a percentage of the total liver section affected. Hepatic inflammation was quantified by counting the number of inflammatory aggregates in 5 non-overlapping fields per rat (at 100× magnification, field of view 4.15 mm^2^) and expressed as the number of aggregates per field. 

#### 2.3.5. Immunohistochemistry

Slices of the medial prefrontal cortex of 16 μM were obtained with a cryostat (Leica CM3050 S). Slices were captured in milliQ water and, the same day, mounted on coated slides (Thermo Scientific, Waltham, MA, USA, Menzel-Gläser Superfrost-Plus). The slides were air-dried overnight and frozen at −20 °C until further use. Before staining, the slides were thawed for 30 min, rehydrated in 1× PBS for 10 min, and thereafter, washed 3 times for 10 min in 1× PBS on the 3D orbital incubation shaker (Stuart gyro rocker SSL3) at 30 rpm. During every incubation step, slides were placed horizontally in a wet chamber (the surface of the slide box was covered with milliQ-soaked tissues). After every upcoming incubation step, slides were washed 3 times for 10 min in 1× PBS on the 3D orbital incubation shaker at 30 rpm. Slides were blocked with 500 μL blocking solution (20% BSA (blocker BSA (10×) in PBS, Thermo Fischer Scientific), 0.2% Triton X-100 (Sigma-Aldrich, St. Louis, MO, USA) in 1xPBS) for 2.5 h. Subsequently, slides were incubated with 500 μL primary antibodies: Anti Iba1, Rabbit, (Wako, 019-19741 (1:500)) and Anti NeuN (Chicken, Millipore, Burlington, MA, USA, ABN91 (1:0000)) diluted in blocking buffer overnight at room temperature. The slides were incubated with 500 μL secondary antibodies: Alexa Fluor^®^ Donkey anti-rabbit 647 (Thermo Fischer Scientific) and Alexa Fluor^®^Goat anti-chicken 555 (Thermo Fischer Scientific), both with 1:200 dilution in 1× PBS for 3 h at room temperature. Thereafter, slides were incubated with 500 μL DAPI (1:1000 diluted in 1% PBS) for 7 min at room temperature, after which slides were individually dipped in milliQ for approximately 5 s to diminish the remaining PBS. After air drying, slides were mounted with 2 drops fluorsave (Calbiochem, Millipore), covered with a cover glass, and placed horizontally in a dry chamber overnight at room temperature to dry. Finally, slides were stored vertically at room temperature until microscope analysis. One image (2 × 2 tile scan, 40× magnification) of the medial prefrontal cortex was obtained on an automated high-content microscope (Leica DMI6000B, Amsterdam, The Netherlands) and analyzed using ImageJ software. The soma cell size area of IBA-1 positive immunoreactive cells was quantified using an average of 5 cells from each rat. 

#### 2.3.6. Gene Expression Analyses

##### RNA Isolation

RNA isolation was conducted using the RNeasy^®^ Mini Kit (Qiagen, Hilden, Germany, cat. nos. 74104 and 74106) and performed according to the manufacturer’s instructions. Samples were checked with NanoDrop 2000/2000c Spectrophotometer (Thermo Fisher Scientific, NanoDrop 2000/2000c software) for concentration and pureness and stored at −20 °C.

##### cDNA Synthesis

The cDNA synthesis was performed according to the manufacturer’s instructions using the SensiFAST™cDNA Synthesis Kit (Bioline, Essex, UK, cat. nos. BIO-65053). The amount of nanograms of mRNA added to the SensiFAST™cDNA Synthesis Kit was equivalent in terms of quantity. The master mix was prepared on ice and mixed by pipetting. Each sample contained total RNA (up to 1 μg), 2 μL; 5× TransAmp Buffer, 4 μL; Reverse Transcriptase, 1 μL, and DNase/RNase free water, up to 20 μL in total. In the thermal cycler, the following program was used: 25 °C for 10 min (primer annealing), 42 °C for 15 min (reverse transcription), and 85 °C for 5 min (inactivation), followed by a 4 °C hold. The cDNA reaction product was stored at −20 °C until further use.

##### qPCR

The qPCR was performed according to the manufacturer’s instructions using the SensiFAST™ SYBR^®^No-ROX Kit (Bioline, cat. nos. BIO-98005). The master mix was prepared, kept away from direct light, and mixed by pipetting. Each sample contained 1× SensiFAST SYBR^®^ No-ROX Mix, 400 nM forward and reverse primer, 2 μL template, and DNase/RNase free water, up to 10 μL in total. Master mixes and templates were pipetted in a Rotor-Disc^®^100 (cat. No. 981311, QIAGEN) by QIAgility pipetting robot (QIAGEN, build 1 (1.6.61) software version 4.14.2) and afterward sealed using the Rotor-Disc Heat Sealer (QIAGEN). qPCR was performed using the Rotor-Gene Q (serial no: 051226, model: 5-Plex HRM, software version: 2.3.1, QIAGEN), with a cycle of 2 min 95 °C for polymerase activation, 40 times 5 s 95 °C for denaturation, 10 s 63 °C for annealing and 10 s 72 °C for extension. The machine kept samples at 4 °C until placed in the 4 °C fridge for storage. Primers Rattus norvegicus Ywhaz, β-actin and Gapdh, SIRT1, and PSD-95 were synthesized by Sigma (for primer pair sequences, see Table 2). For the statistical analyses, three reference genes were chosen in the first instance (Ywhaz, β-actin, and Gapdh). However, since Gapdh expression may be influenced by diet, it was added as an exploratory gene. Therefore, gene expression was normalized to the Ywhaz/β-actin ratio, and results were subjected to two-way ANOVA for normally distributed variables with equal variances.

#### 2.3.7. Hippocampal Insulin Levels

Isolated Hippocampi from ZDF rats, stored at −80 °C, were weighed before extracting total protein. Tissues were first homogenized by sonication in 10× volume of Tissue Lysis Buffer (N-PER-Halt) supplemented with protease inhibitor cocktails (Thermo Fisher). Tissue lysis was centrifuged at 12,000× *g* for 10 min at 4 °C, and the supernatant was transferred to a new Eppendorf tube. Total protein concentration was determined by the BCA method (Thermo Scientific™ Pierce™ 96-Well Plates, USA). Standard curves of BSA using a range of concentrations between 250 and 2000 µg/mL were used in order to determine the tissue protein concentrations. For ELISA, 50 µL of tissue supernatants were used in a 96-well antibody-coated microplate. Two sets of positive control samples were included in each ELISA: Rat insulin standard (supplied in the Ultra Sensitive Kit, CrystalChem, High-Performance Assay, Elk Grove Village, IL, USA). Low range assay (0.1–6.4 ng/mL) of insulin was used. Results were presented as µg/mL wet tissue.

### 2.4. Statistical Analysis

All analyses were executed using GraphPad Prism version 9 (Graphpad Software, San Diego, CA, USA) and IBM SPSS Statistics (Version 27; IBMCorp, Armonk, NY, USA). Graphs were made in Graphpad Prism. The main effect of sex, genotype (for ZDF), or feeding regime (for LE rats) on subsequent or single measurements were analyzed using either repeated measures ANOVA using a mixed-effects model or were subjected to two-way ANOVA for normally distributed variables with equal variances. A *p*-value of ≤0.05 was considered statistically significant. The Greenhouse–Geisser correction was used where appropriate for violation of sphericity. Significant main and interaction effects were followed by Tukey’s post hoc analyses. 

## 3. Results

### 3.1. Effects of the Two Dietary Regimes in Healthy Long Evans (LE) Rats on Behavior and Physiological Outcomes

A detailed statistical description and graphic illustration of the comparison between two groups of LE rats (LE sucrose versus LE grain) are presented in full in the Appendix A. In sum, the ad libitum feeding and sucrose exposure during operant training resulted in a higher body weight compared to male LE grain rats, whereas no difference was observed between female LE groups (Appendix A). Performance during the operant training stages prior to the VD task showed that LE-sucrose rats needed more training sessions in all training stages (Appendix A). During the VD task itself, the LE-sucrose group showed reduced performance during the first 10 sessions (Appendix A), but a comparable level of performance was seen once rats reached the VD criteria of >70% correct (Appendix A). The number of sessions to reach the criterion did not differ between groups (Appendix A). The number of sessions required during FR stages was enhanced in LE-sucrose rats (Appendix A), while the motivation during PR was not different between groups (Appendix A). Blood analyses revealed that LE-sucrose rats had significantly increased glucose levels (Appendix A) and increased total cholesterol and triglycerides levels (Appendix A), but insulin levels were not altered (Appendix A). No effects of the dietary regime were observed on the relative weight of perigonadal and mesenteric white adipose tissue and the relative liver weight (Appendix A–C). However, the number of immune cell aggregates within the liver was significantly increased in female LE-sucrose rats (Appendix A). Interestingly, in LE-sucrose rats, we observed a significant decrease in the number of both NeuN and IBA-1 positive cells in the PFC compared to LE-grain rats (Appendix A–C), but no differences in the IBA-1 immunoreactive cells area (Appendix A). No significant group differences were observed in SIRT1, PSD-95, and GAPDH mRNA expression levels in the PFC (Appendix A). 

### 3.2. Body Weight Development, Locomotor Activity, and Ad Libitum Food Intake in ZDF Rats

Figure 1A presents the enhanced body weight gain in obese versus lean ZDF rats from weeks 4 to 14 of age (Fgenotype(1, 32) = 132.6, *p* < 0.0001). A significant interaction effect between genotype and sex was found (Fgenotype x sex(1, 32) = 9.184, *p* = 0.0048), which indicated that the difference in body weight development between the female groups was larger (46.09% at week 14) compared to the difference between the male groups (21.40% at week 14) (Figure 1B). Post hoc testing per gender indicated that significant body weight differences emerged from week 6 (*p* < 0.05) for females and week 7 for males (*p* < 0.05). Average locomotor activity was significantly lower in obese compared to lean ZDF rats, both when assessed during 48h sessions in LABORAS (Fgenotype(1, 30) = 100.2, *p* < 0.0001) (Figure 1C) and during the 10 first sessions of the VD task (Fgenotype(1, 27) = 16.41, *p* = 0.0004) (Figure 1D). Moreover, the difference in locomotor activity between obese and lean ZDF rats in females was larger compared to males in LABORAS (Fsex(1, 30) = 26.34, *p* < 0.0001, Fgenotype x sex(1, 30) = 22.08, *p* < 0.0001). Food intake (corrected for body weight) decreased over-development (Fweeks(1, 51) = 82.32, *p* < 0.0001) and showed a significant interaction effect for genotype over weeks (Fgenotype(1, 318) = 14.91 *p* = 0.0001; Fweeksxgenotype(9, 318) = 7.65, *p* = 0.001), but without an effect of sex. Post hoc analyses per week for genotype comparisons (including both sexes) indicated that obese ZDF rats consumed more food during ad libitum access on weekends at weeks 5, 6, and 8 weeks of age (Appendix A). Overall, no significant effect of genotype on the duration spent on ad libitum eating and drinking during the 48 h sessions in LABORAS was observed (Appendix A).

### 3.3. Cognitive Performance and Level of Motivation in ZDF Rats

Obese rats needed fewer training sessions compared to lean ZDF rats during the training stages of autoshaping (Fgenotype(1, 31) = 10.57 *p* < 0.01), punish incorrect (Fgenotype(1, 32) = 9.17, *p* < 0.01) and moving punish incorrect (Fgenotype(1, 32) = 15.54, *p* < 0.001). Only during must touch training do obese ZDF rats require more training sessions (Fgenotype(1, 32) = 4.35, *p* < 0.05) (Appendix A). Compared to males, females needed more sessions to reach the criteria on punish incorrect (Fsex(1, 32) = 4.53, *p* = 0.041; Fgenotype x sex(1, 32) = 4.37, *p* = 0.045) and the moving-punish-incorrect training stage (Fsex(1, 32) = 12.01, *p* < 0.01; Fgenotype x sex(1, 32) = 12.79, *p* < 0.01). The analyses across the first 10 VD sessions revealed that obese rats showed an enhanced learning capacity compared to lean ZDF rats (Fgenotype(1, 32) = 35.12, *p* < 0.0001; F(9, 285)genotype x session = 8.263, *p* < 0.0001). No significant effect of sex was observed (Fsex(1, 32) = 2.448, *p* = 0.1275) (Figure 2A). When averaging the correct responses over the first 10 sessions, it was observed that obese ZDF rats showed a higher percentage of correct responses (Fgenotype(1, 32) = 37.00, *p* < 0.0001) (Figure 2B). Once the VD criterium was reached by each individual rat, obese ZDF rats still showed a significantly higher percentage of correct responses compared to lean ZDF rats (Fgenotype(1, 27) = 12.97, *p* < 0.01) (Figure 2C). Obese rats required fewer sessions to reach the VD criterium (Fgenotype(1, 31) = 28.28, *p* < 0.001), and it was observed that, in particular, female lean ZDF rats required the highest number of sessions to reach the VD criterium (F(1, 31)genotype x sex = 5.99, *p* = 0.020; Fsex(1, 31) = 6.49, *p* = 0.016) (Figure 2D). When the motivation to respond was subsequently assessed, no genotype differences during FR stages (Figure 2E) and the PR schedule of reinforcement were observed (Figure 2F). When altering the reward pellet during PR sessions, we observed no effect of genotype or reward pellet type on the average number of presses for the new sucrose reward during PR sessions 6, 8, and 10 versus the average number of presses for the known grain reward on session 5, 7, 9 (Figure 2G).

### 3.4. Blood Glucose, Insulin, and Lipids in ZDF Rats

Fasted glucose levels varied over weeks of age (Fweeks(3, 84) = 5.96, *p* < 0.005) and were significantly higher in obese rats compared to lean ZDF rats (Fgenotype(1, 32) = 13.03, *p* = 0.001), without a difference in sex (Fsex(1, 32) = 1.80, *p* = n.s.) (Figure 3A). Fasted insulin levels increased over weeks of age in the male and female obese group while the insulin levels stayed stable in lean ZDF rats (Fgenotype x weeks(4, 123) = 13.49, *p* < 0.0001; (Fgenotype(1, 32) = 45.47, *p* < 0.0001) (Figure 3C). Interestingly, the increment in insulin levels over weeks was higher in obese male rats compared to obese female rats (Fgenotype x sex x weeks(4, 123) = 4.74, *p* < 0.05; (Fgenotype x sex(1, 32) = 7.29, *p* < 0.05). The average glucose and insulin values of the five time points (weeks 4–12 of age) are shown in Figure 3B–D. ZDF obese rats had higher total cholesterol (Fgenotype(1, 32) = 122.6, *p* < 0.0001), triglycerides (Fgenotype(1, 32) = 21.48, *p* < 0.0001) as well as high-density lipoprotein levels (Fgenotype(1, 16) = 53.70, *p* < 0.0001) compared to lean rats (Figure 4A–C). Pooled plasma from ZDF obese and lean rats was investigated for lipoprotein fractionation and showed that obese rats mainly had enhanced high-density lipoprotein fractions in cholesterol and very low-density lipoprotein fractions in triglycerides, compared to lean rats (Figure 4D,E).

### 3.5. Histological Analyses of White Adipose Tissues and Liver in ZDF Rats

The relative adipose tissue weights and adipocyte sizes were higher in obese compared to lean ZDF rats. This was the case for both the relative weight of perigonadal (Fgenotype(1, 32) = 219.4, *p* < 0.0001) and mesenteric white adipose tissue (Fgenotype(1, 32) = 218.5, *p* < 0.0001) (Figure 5A,B), and the adipocyte sizes (Fgenotype(1, 15) = 72.49, *p* < 0.0001; Fgenotype(1, 16) = 73.70, *p* < 0.0001), respectively (Figure 5C,D). Representative images of perigonadal and mesenteric white adipose tissue are presented in Figure 5E,F. The distributions in the percentage of total cells per cell size are presented in Figure 5G,H. The pathological investigation of the liver (Figure 6A) revealed starting hepatic microvascular steatosis (Fgenotype(1, 31) = 62.23, *p* < 0.0001) and hypertrophy (Fgenotype(1, 31) = 39.88, *p* < 0.0001) in obese rats compared to lean ZDF rats (Figure 6D). Furthermore, significantly higher relative liver weights were found in obese rats (Fgenotype(1, 32) = 70.05, *p* < 0.0001) (Figure 6B), and the number of immune cell aggregates was significantly increased in obese compared to lean ZDF rats (Fgenotype(1, 31) = 20.12, *p* < 0.0001) (Figure 6C).

### 3.6. mPFC NeuN and IBA-1 Positive Cells, mPFC SIRT1 and PSD-95 mRNA Expression, and Hippocampal Insulin Levels in ZDF Rats

No significant differences between ZDF obese and lean rats or differences between males and females were found for the number of NeuN (Fgenotype(1, 16) = 0.35, *p* = n.s.; Fsex(1, 16) = 0.01, *p* = n.s.) and IBA-1 positive cells (Fgenotype(1, 17) = 0.44, *p* = n.s.; Fsex(1, 17) = 0.77, *p* = n.s.) (Figure 7A–C). Furthermore, no significant effects of genotype and sex were found for IBA-1 immunoreactive soma cell size (Fgenotype(1, 15) = 0.224, *p* = n.s.; Fsex(1, 15) = 0.7203, *p* = n.s.) (Figure 7D). The mRNA expression of SIRT1, PSD-95 and GAPDH was not different between groups either (SIRT1: Fgenotype(1, 30) = 0.2086, *p* = n.s., Fsex(1, 30) = 0.5694, *p* = n.s.; PSD-95: Fgenotype(1, 30) = 0.2167, *p* = n.s., Fsex(1, 30) = 1.351, *p* = n.s.; GAPDH: Fgenotype(1, 31)= 3.489, *p* = n.s., Fsex(1, 30) = 2.412, *p* = n.s.) (Figure 8A–C). The level of brain insulin was measured in the hippocampus and was not different between ZDF obese and lean rats or between sexes (Fgenotype(1, 30) = 0.045, *p* = n.s., Fsex(1, 30) = 0.047, *p* = n.s.). 

## 4. Discussion

This study investigated whether cognitive performance is impacted during adolescence in a genetic rat model of obesity and diabetes. Interestingly, we observed that adolescent ZDF obese rats cognitively outperformed both ZDF lean rats and healthy LE rats implemented with the same food regime. This enhanced cognition lasted during the entire visual discrimination period. Clear features of type 2 diabetes were observed in ZDF obese rats from week 6 of age onwards. Furthermore, early adult ZDF obese rats presented mild signs of hepatic and white adipose tissue inflammation. Interestingly, no alterations in the number and morphology of neuron and microglia cells, nor differences in PSD-95, SIRT-1, and GAPDH expression, were observed between young adult ZDF obese and lean rats. The comparison between the LE-sucrose and LE-grain groups showed that the LE-sucrose group had a slower acquisition rate during the pre-training stages and a lower cognitive performance during the first ten VD sessions. However, sucrose exposure did not impact the final cognitive performance and neither impact motivation levels. Sucrose exposure resulted in enhanced blood glucose, total cholesterol, and triglycerides levels and lower levels of NeuN and IBA-1 positive cells in the PFC, compared to LE-grain rats.

The cognitive superiority of the ZDF obese rats was unexpected because previous studies reported either no changes in learning [34] or spatial memory deficits [13,14,35]. Since the enhanced performance was present in both the visual discrimination task and most of the pre-training acquisition phases, it is hard to interpret whether the ZDF obese rats have an enhanced cognition or a different learning capacity. A time response study using a diet-induced obesity model revealed that insulin resistance appeared from the 7th week of fructose feeding, whereas cognitive dysfunction (memory function in Morris Water Maze) appears only after the 20th week of fructose exposure [14]. Comparable results were found using Zucker rats, in that cognitive alterations in the passive avoidance test (lower capacity to learn and to keep away from unpleasant electric shock) were observed in 20-week-old obese Zucker rats, with no differences in 12- and 16-week-old, compared to lean rats [36]. Moreover, rat strains with a genetic predisposition to obesity did not differ in their ability to experience reward (in a condition place preference task) [37]. It would have been worthwhile to investigate in the current used touchscreen higher-order cognitive task whether older adult ZDF obese rats show cognitive dysfunction. However, this might challenge ethical decisions since we clearly observed that when obese ZDF rats age, diabetic disease complications start to emerge. In line with the literature, in some animals, we observed progressive nephropathy, increasing proteinuria resulting in chronic renal insufficiency, and impaired wound healing in early adulthood [38,39]. 

It could be that the hyper-insulinemic condition of ZDF obese rats influenced cognitive performance. In support, we found a negative correlation between the number of sessions in the visual discrimination task and insulin concentration in the blood (r = −0.4801, *p* < 0.0035), indicating a faster learning acquisition when insulin levels are high. Since insulin can cross the blood-brain-barrier, it is possible that hyperinsulinemia could stimulate cognitive performance via neurogenesis pathways and/or play an adaptative protective role in adolescent ZDF obese rats by directly suppressing pro-inflammatory cytokines and preventing neuronal impairment [17,40]. However, we did not investigate the causal relationship between insulin and cognitive performance. This requires future investigations employing local brain infusions of insulin or a systemic injection with insulin to assess the direct brain effects. We did, however, measure the brain insulin levels in the hippocampus but did not observe a difference between ZDF obese versus lean rats when measured at 14–16 of age. Growing evidence supports the role of SIRT1 in the regulation of insulin sensitivity [41] and neuroinflammatory responses [42]. Additionally, a decrement of SIRT1 in the brain is associated with cognitive impairment [43]. Remarkably, SIRT1 induced cognitive enhancement even in healthy non-transgenic mice and increased neuronal plasticity proteins such as PSD-95 [44]. Our results do not depict significant differences in the expression of SIRT1 and PDS-95 in the brain of ZDF obese rats compared to lean rats. Furthermore, no significant differences were observed in the number of neurons and microglia or the morphology of the microglia soma sizes, which is in line with a previous study that reported no alteration in the morphology of astrocytes and microglia in ZDF obese versus ZDF lean rats [45]. Therefore, it is unlikely that SIRT1, PDS-95, or the number of neurons and microglia can explain the enhancement of cognitive performances in ZDF obese rats. Previous time response studies showed indeed that a decrease of neuron morphological changes of microglia with an increase of the area of soma started to appear in 20-week-old obese Zucker rats (and not at 12- and 16-week-old) [36] and 30-week-old ZDF rats (and not at 12 and 20 weeks old) [46]. A reduction in brain cholinergic (VAChT and α7nAChR) and synaptic markers (synaptophysin and synaptic vesicle glycoprotein 2B) have been observed in the frontal cortex and hippocampus in obese Zucker rats, but these reductions only started to emerge at 20 weeks of age and were not present at 12 and 16 weeks of age [47]. Since we performed our brain histology and mRNA expression experiments on rats who were 14–16 weeks of age, this supports our idea that negative effects due to the obesity phenotype have not yet occurred within the prefrontal cortex. 

As expected, ZDF obese rats developed obesity as measured by increased body weight, liver weight, and perigonadal- and mesenteric white adipose tissue weights. Given the hyperphagic nature of leptin resistance in genetic animal models, we assessed whether the enhanced cognitive performance could be explained by an increased motivation to obtain reward. We observed no group differences between ZDF obese and lean rats in early adulthood in PR sessions, nor toward a new type of reward pellets in the PR test. Zucker rats of a comparable age also showed no differences with lean rats during PR sessions where a relatively high effort was required [37], but when lower effort was required during PR and FR lever press training, a higher motivated behavior was observed [37,48]. The interplay between leptin, dopamine, reward, and motivation entails a complex physiological network. Leptin may act at discrete brain loci and use diverse molecular signaling pathways to influence striatal dopamine release [49]. Since ZDF obese rats have a mutation in the leptin receptor, leptin signaling via this receptor is, per definition, hampered. This confounds the relevance and interpretability of biological changes in leptin concentrations and, thus, also possible conclusions from a comparison of leptin levels between ZDF lean and obese rats. The control and regulation of the feeling of satiety is, in our view, different from the feeling of motivation. Still, we cannot fully exclude the possibility that motivational differences between ZDF lean and obese rats played a role in the observed differences in visual discrimination performance. 

A potentially confounding factor of this study is that we applied social isolation in order to control food intake per animal. Previous studies have consistently demonstrated that early social isolation results in anxiety-like behavior and cognitive impairment in adulthood [50]. Importantly, impairments in impulsive action, decision making, and cognitive flexibility are most often observed under novel or challenging circumstances [51,52]. Moreover, a reduced dopamine modulation of prefrontal cortex function due to social isolation has been observed in several investigations [51], as well as reductions in inhibitory synapses in the PFC [52,53]. Still, this potential source of stress was present for all groups of rats tested in our study. Nevertheless, it may be possible that interaction effects occur between the consequences due to social isolation and the ZDF genotype on cognition.

We clearly observed in our locomotor activity data, but also based on video recording during operant testing, that ZDF lean rats, mainly females, were hyperactive. Comparable findings of enhanced locomotor activity have been observed previously in Obese Resistant (OR) rats [37]. Due to the extraordinary hyperactive behavior (high arousal) of the ZDF lean rats, which may subsequently have consequences for less sustained attention to perform and learn the visual discrimination task, we had concerns about whether the ZDF lean group was the correct ‘control group’ for the ZDF obese rats. Therefore, we included the group of LE rats receiving the same feeding regime (LE-grain) as both the ZDF groups in the analyses of the VD task. Based on the analyses described in Appendix A, it can be concluded that ZDF obese rats also outperformed LE grain rats. Thus, apparently, the ZDF obese rats show a truly higher cognitive performance compared to these two control groups. Hence, the extraordinary hyperactive behavior of the ZDF lean rats, which was not observed in LE rats, cannot explain the significantly enhanced cognitive performance of the ZDF obese rats. Importantly, a relation between a lower locomotor activity level and improved cognitive performance could exist. Potentially, a lower locomotor activity level may induce a higher level of (sustained) attention and thereby improve cognition. 

Although the genetic ZDF model is a very different disease model compared to diet-induced obesity, the fact that exactly the same experiments were performed in both the LE and ZDF groups allows the valuable possibility to gain insight into the consequences of both models on the same higher-order cognitive performance during development. We observed that in LE rats, food restriction in association with grain pellet reward significantly speeds up the acquisition of a complex cognitive task relative to ad libitum feeding and sucrose pellet as a reward. It has previously been suggested that ad libitum feeding by itself leads to metabolic morbidity, with negative effects on various bodily functions, including cognition [54]. Furthermore, it has been demonstrated that adolescent sucrose exposure in animals reduces cognitive functioning [55]. Blood glucose cholesterol and triglyceride levels were enhanced in LE-sucrose rats compared to LE-grain rats, as was the infiltration of inflammatory cells in the liver. The number of microglia and neurons in the prefrontal cortex was decreased in LE-sucrose rats, which typically has been associated with reduced cognitive performance [56].

## 5. Conclusions

Taken all data together, ZDF obese rats showed an increased cognitive performance at adolescence despite several physiological symptoms of obesity and type 2 diabetes. In early adulthood, an onset of inflammation in the liver was present, while no brain alterations were detected. We propose the possibility that prefrontal function during the induction of obesity and diabetes in a genetic ZDF rodent model was maintained by hyperinsulinism and even resulted in improved cognitive performance. Future research could explore this potential role of insulin further by investigating, for example, the effects of central insulin administration on cognitive performance or the assessment of insulin signaling in adolescent ZDF obese rats.

## Figures and Tables

**Figure 1 cells-12-02463-f001:**
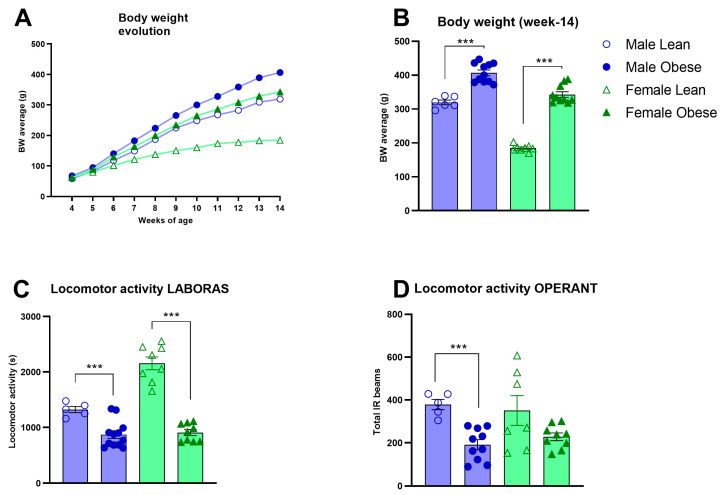
Body weight and locomotor activity in obese and lean ZDF rats. (**A**) Body weight development during adolescence and early adulthood between weeks 4 and 14 of age. (**B**) Bodyweight comparison at week 14. Values of (**A**,**B**) represent the mean (±SEM). (**C**) Locomotor activity in Laboras homecages (seconds). Values in C represent the mean of 5 sessions of 48h each in the Laboras homecage ± SEM. (**D**) Average locomotor activity during the first 10 sessions of the visual discrimination stage of operant testing. Values represent the average number of infrared beam crosses during a session ± SEM. Asterisks indicate lean vs. obese group comparisons per gender ***: *p* < 0.001.

**Figure 2 cells-12-02463-f002:**
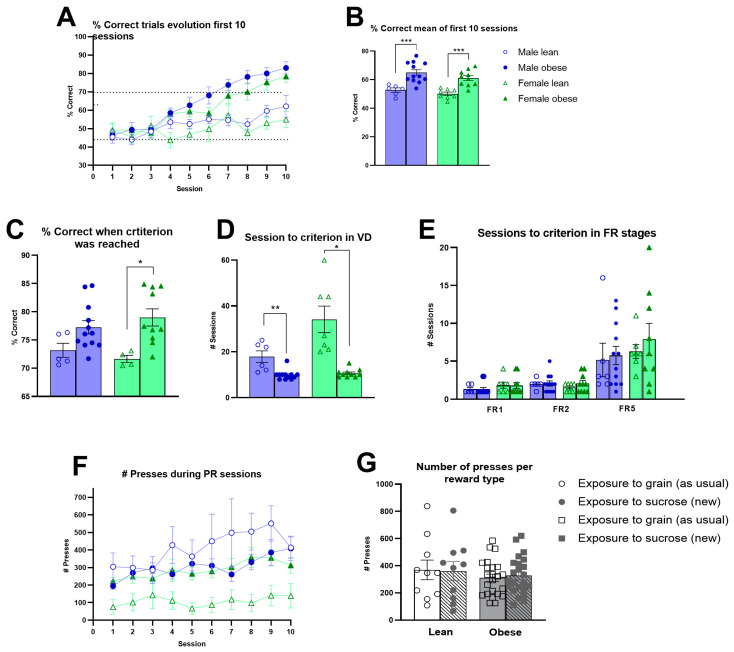
Cognitive performance and level of motivation in obese and lean ZDF rats. (**A**) The evolution of the percentage of correct trials during the first 10 sessions of the visual discrimination (VD) task. (**B**) Average values of the percentages of the correct trials during the first 10 VD sessions. (**C**) Average percentage of correct trials of the three consecutive trials when task criterion was reached. (**D**) Number of sessions required to reach criterion during the VD task. Values during the VD task represent the group mean (±SEM). (**E**) Number of sessions required to reach criterion in different fixed ratio phases in male and female lean vs. obese rats. (**F**) Average group level of active responses to obtain a reward during the 10 sessions of the progressive ratio in male and female lean vs. obese rats. (**G**) Effects of the reward type (sucrose vs. grain pellets) on the motivation to perform active responses to obtain the reward. Values during the FR and PR sessions represent the group mean (±SEM). Asterisks indicate lean vs. obese group comparisons per gender ***: *p* < 0.001, **: *p* < 0.01, *: *p* < 0.05.

**Figure 3 cells-12-02463-f003:**
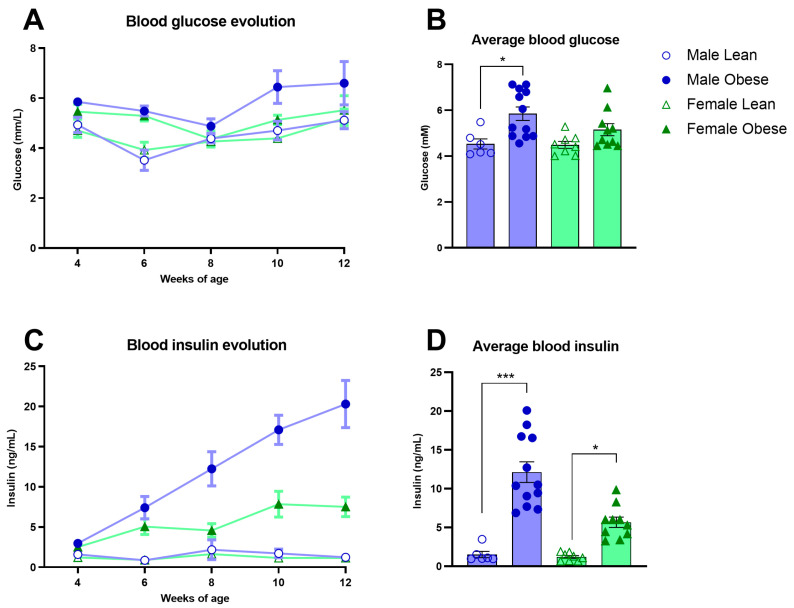
Blood glucose and plasma insulin levels in obese and lean ZDF rats. (**A**,**C**) Fasted glucose and insulin levels over time at weeks 4, 6, 8, 10, and 12 of age. (**B**,**D**) Average fasted glucose and insulin levels of the measured five time points. Values represent the group mean (±SEM). Asterisks indicate lean vs. obese group comparisons per gender ***: *p* < 0.001, *: *p* < 0.05.

**Figure 4 cells-12-02463-f004:**
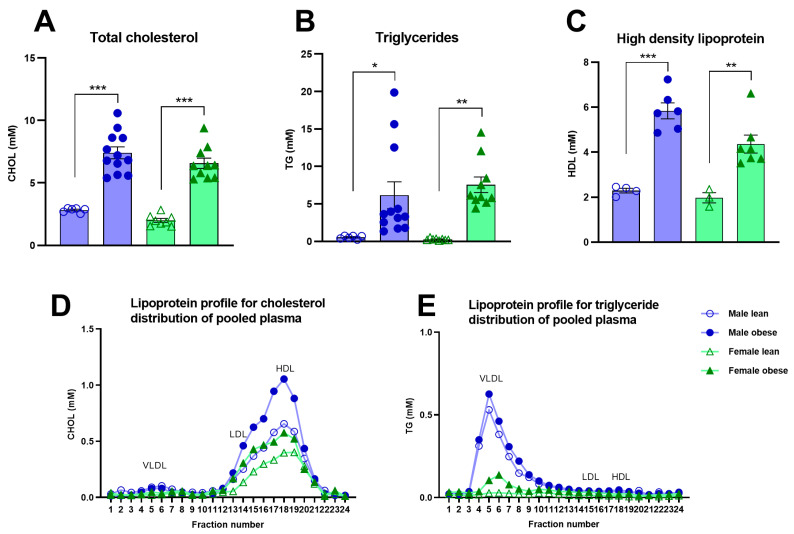
Blood total cholesterol, triglycerides, and high-density lipoprotein levels in obese and lean ZDF rats. Average total cholesterol (**A**), triglycerides (**B**), and high-density lipoprotein (**C**) at week 12 of age. Lipoprotein profiles of pooled plasma at week 12 of age with the physiological distribution of cholesterol (**D**) and triglycerides (**E**) among lipoproteins. Twenty-four fractions were collected and further assessed for cholesterol and triglyceride concentrations. Fractions 4 to 7 were determined as very low-density lipoprotein, 11 to 16 as low-density lipoprotein, and 17 to 25 as high-density lipoprotein. Values represent the group mean (±SEM). VLDL = very low-density lipoprotein; LDL = low-density lipoprotein; HDL = high-density lipoprotein. Asterisks indicate lean vs. obese group comparisons per gender ***: *p* < 0.001, **: *p* < 0.01, *: *p* < 0.05.

**Figure 5 cells-12-02463-f005:**
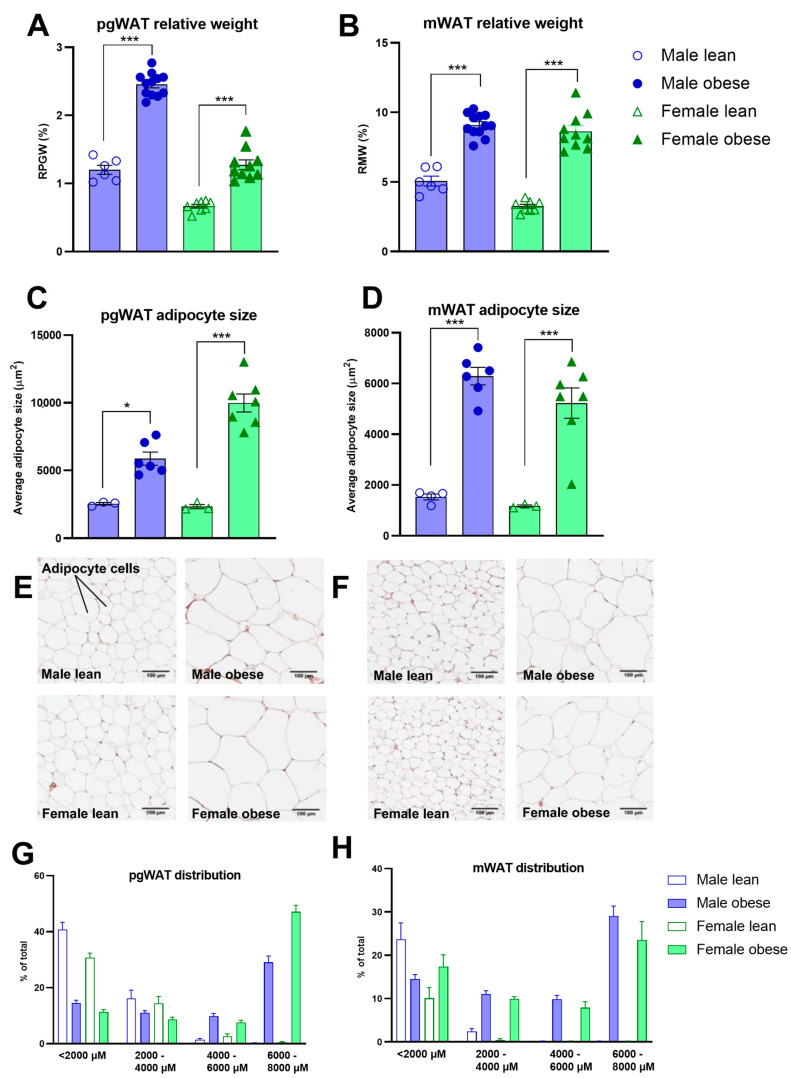
Histological profiles of white adipose tissue in obese and lean ZDF rats. (**A**,**B**) Average relative weights of perigonadal (pgWAT) and mesenteric (mWAT) white adipose tissues. (**C**,**D**) pgWAT and mWAT adipocyte size and their percentage distribution per cell size (**G**,**H**) in both lean and obese males and females. (**E**,**F**) Representative images of pgWAT and mWAT tissues. Values represent the mean group (±SEM). Asterisks indicate lean vs. obese group comparisons per gender ***: *p* < 0.001, *: *p* < 0.05.

**Figure 6 cells-12-02463-f006:**
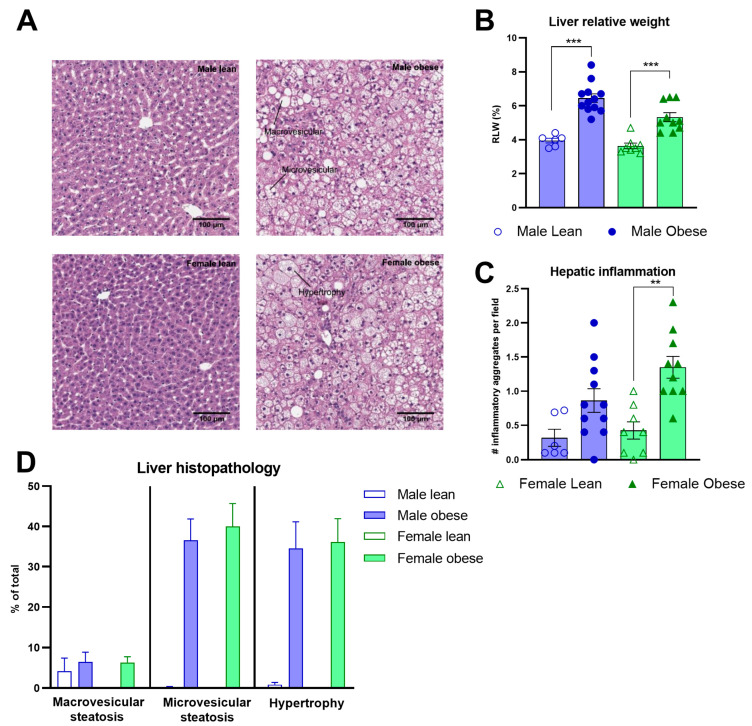
Liver histopathology in obese and lean ZDF rats. (**A**) Representative images of the liver H&E staining showing the histopathological affections in ZDF obese rats. (**B**) Relative liver weights. (**C**) Hepatic inflammation is expressed as the number of observed aggregates per field. (**D**) The average percentage of liver tissue steatosis (macro- and microvesicular steatosis) and hypertrophy. Values represent the group mean (±SEM). Asterisks indicate lean vs. obese group comparisons per gender ***: *p* < 0.001, **: *p* < 0.01.

**Figure 7 cells-12-02463-f007:**
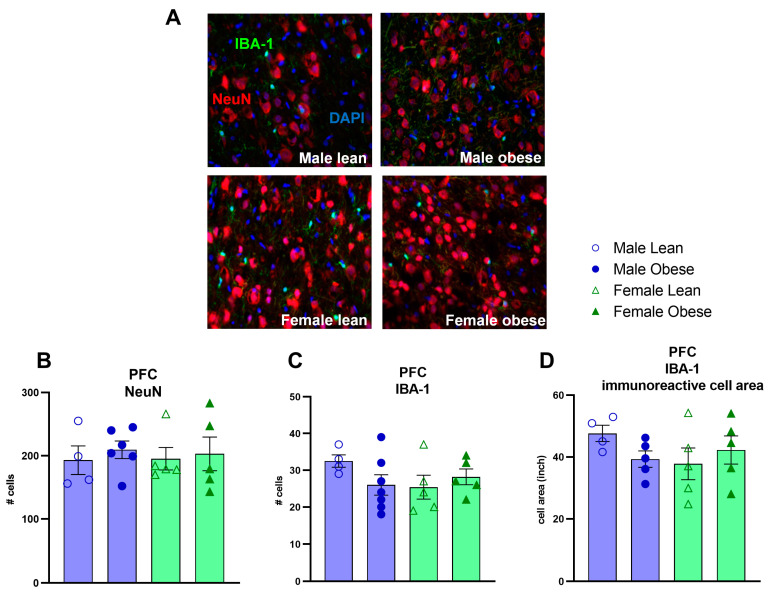
Evaluation of NeuN, IBA-1 positive cells, and IBA-1 immunoreactive soma cell size area by immunohistochemistry in the medial prefrontal cortex in obese and lean ZDF rats. (**A**) Representative images of NeuN and IBA-1 immunoreactive cells. (**B**) The number of NeuN-positive cells. (**C**) Number of IBA-1 positive cells. (**D**) IBA-1 immunoreactive soma cell size. Values represent the group mean (±SEM).

**Figure 8 cells-12-02463-f008:**
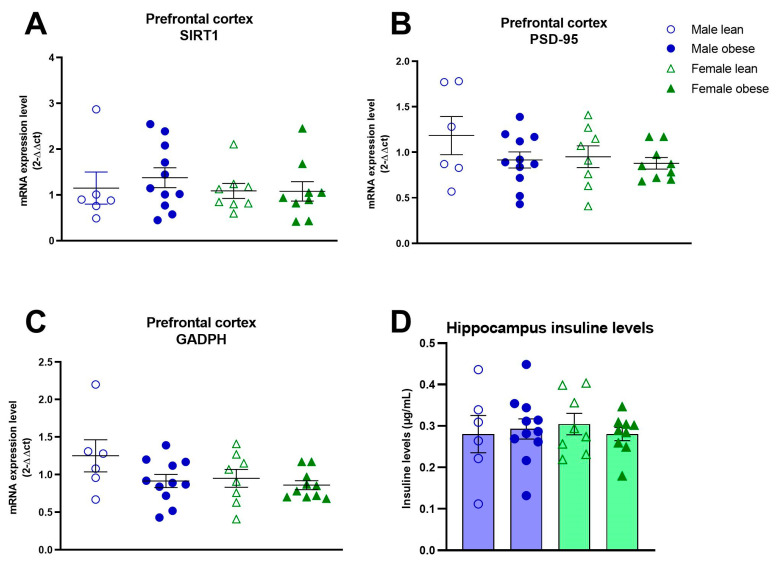
mRNA expression levels in the medial prefrontal cortex and hippocampal insulin levels in obese and lean ZDF rats. mRNA levels of SIRT1 (**A**), PSD-95 (**B**), and GAPDH (**C**). Hippocampal insulin levels (**D**). Values represent the group mean (±SEM).

**Table 1 cells-12-02463-t001:** Different criteria during cognitive testing.

Learning Stage	Criteria	Description
Habituation	Complete one session	The rat was provided with 25 pellets at variable time intervals.
Autoshaping	All 25 pellets eaten	In order to learn the association between the stimulus and the reward, the rat received a reward regardless of whether it touched the stimulus and would receive the reward immediately when it touched the image.
Must touch	In two sessions, 25 correct responses and all 25 pellets eaten	The rat had to touch the stimulus to receive a reward; otherwise, no reward was given.
Punish incorrect	At least 70% correct responses for two sessions (Nr correct/Nr correct + Nr incorrect) × 100	The rat receives negative feedback (house light on) when it touches the screen in the wrong location. Then, no pellet is given.
Moving punish incorrect	At least 70% correct responses for two sessions (Nr correct/Nr correct + Nr incorrect) × 100	Similar to the punish incorrect, but the stimulus changes position from left to right in a pseudorandom order.
Visual discrimination	At least 10 sessions and anaverage of 70% correct responses across three consecutive sessions (Nr correct/Nrcorrect + Nr incorrect) × 100	Similar to moving punish incorrect, however, the circular stimulus is replaced by two striped images. The designated correct stimulus (counterbalanced between rats) changes position in a pseudorandom order.
Fixed Ratio 1	50 correct responses	The rat had to press once to receive a reward.
Fixed Ratio 2	100 correct responses	The rat had to press twice to receive a reward.
Fixed Ratio 5	250 correct responses	The rat had to press five times to receive the reward.
Progressive Ratio	Complete 10 (ZDF rats) or 20 sessions (LE rats)	On each subsequent trial, the reward response requirement increased on a linear basis (+4).

**Table 2 cells-12-02463-t002:** Primers used in the qPCR analysis.

Name	Sequence 5′ -> 3′	Tm	GC%
YWHAZ	fw: GACAAGAAAGGAATTGTGGACCAGT rv: GGGCCAGACCCAGTCTGATG	61.44 62.26	44.00 65.00
GAPDH	fw: CCACCAACTGCTTAGCCCCC rv: TGGTCATGAGCCCTTCCACG	63.12 62.18	65.00 60.00
SIRT1	fw: AGATACCTTGGAGCAGGTTGCAG rv: AGATGCTGTTGCAAAGGAACCATGA	62.51 63.48	52.17 44.00
β-actin	fw: CGTGAAAAGATGACCCAGATCA rv: AGAGGCATACAGGGACAACAC	58.40 59.72	45.45 52.38

## Data Availability

The datasets used and/or analyzed during the current study are available from the corresponding author upon reasonable request.

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
