# Peer review of "Cognitive Performance during the Development of Diabetes in the Zucker Diabetic Fatty Rat"

_cells, 2023, doi:10.3390/cells12202463_

Round 1
Reviewer 1 Report
This is an interesting and timely study. The following issues should be addressed.
1. The abstract refers to three groups but only two seem shown in the figures. That needs to be corrected.
2. One wonders if lower activity levels might have contributed to improved performance in the cognitive test. The discussion about this is not convincing, also because not all three groups are shown.
3. The limitation of using only one cognitive test should be acknowledged. Comparing these results with water maze or passive avoidance data seems hard.
Reviewer 2 Report
The manuscript authored by Spoelder et al. delves into the influence of obesity and type 2 diabetes on cognitive function in adolescent rats. The study's outcomes reveal that adolescent rats with obesity and diabetes outperformed their counterparts in visual discrimination tasks. Insulin levels in ZDF mice exhibited a significant increase over time. Additionally, early indicators of liver issues and enlarged adipocytes emerged in these rats during early adulthood. Intriguingly, no notable distinctions were found in neuron and microglia counts or gene expression levels in the prefrontal cortex. This suggests that despite their early diabetic condition, adolescent obese rats may exhibit enhanced cognition. While the meticulously conducted experiments yield novel and scientifically significant findings, certain aspects require further attention and correction before publication.
Comments:
1. I recommend that the authors enhance the clarity of their graphical representation when presenting the statistical differences from multicomparison tests. Currently, none of the graphs explicitly indicate whether there are significant posthoc test variations between groups and, if so, what those variations are. In the Methods and Materials section, the authors mention the use of mixed-model or two-way ANOVA with the Turkey posthoc test, but it would be more informative if this were explicitly specified in the figure legends. Some figure legends also require further refinement, such as Fig S1, where the detailed reporting of statistical test results is provided, but it can be challenging to discern which results correspond to panels A, B, C, or D.
2. In this study, the authors chose to individually house mice starting at around 4 weeks of age. While this decision improved certain aspects of the study, such as facilitating the measurement of food intake, it also comes with a drawback. Social isolation during adolescence is a potent source of chronic stress, leading to elevated plasma corticosterone levels and impaired cognitive function (source: https://doi.org/10.1016/j.physbeh.2021.113440). Given that this study has no group-housed comparison group, the authors must address the potential effects of social isolation.
3. The authors assert that, "A higher acquisition rate, as observed in our ZDF obese rats during pre-training sessions for VD and during FR training, should not be considered a true indicator of motivation. Consequently, speculating that the improved cognitive performance of ZDF obese rats is solely motivated by the anticipation of receiving caloric reward pellets is not substantiated." Given the significance of this aspect in the overall study, the presented results have not entirely convinced me that the superior performance of obese rats is unrelated to the motivation stemming from the prospect of receiving caloric rewards upon correct responses. Is it feasible to assess leptin levels concurrently with the measurement of insulin levels, using the same samples that were previously collected?
Round 2
Reviewer 2 Report
Authors addressed all my comments